# Two-dimensional vacancy platelets as precursors for basal dislocation loops in hexagonal zirconium

Si-Mian Liu [1], Irene J. Beyerlein [2] & Wei-Zhong Han [1✉]

Zirconium alloys are widely used structural materials of choice in the nuclear industry due to their exceptional radiation and corrosion resistance. However long-time exposure to irradiation eventually results in undesirable shape changes, irradiation growth, that limit the service life of the component. Crystal defects called <c> loops, routinely seen no smaller than 13 nm in diameter, are the source of the problem. How they form remains a matter of debate. Here, using transmission electron microscopy, we reveal the existence of a novel defect, nanoscale triangle-shaped vacancy plates. Energy considerations suggest that the collapse of the atomically thick triangle-shaped vacancy platelets can directly produce <c> dislocation loops. This mechanism agrees with experiment and implies a characteristic incubation period for the formation of <c> dislocation loops in zirconium alloys.

[1] Center for Advancing Materials Performance from the Nanoscale, State Key Laboratory for Mechanical Behavior of Materials, Xi'an Jiaotong University, 710049 Xi'an, China. [2] Mechanical Engineering Department, Materials Department, University of California, Santa Barbara, CA 93106-5070, USA. ✉email: wzhanxjtu@mail.xjtu.edu.cn

Zirconium (Zr) alloys are widely used for fuel cladding tubes and pressure vessels in nuclear reactors. The anisotropic dimension changes that occur in these alloys under irradiation and no externally applied stress are known as irradiation growth, a common and undesirable phenomenon. With demands relentlessly increasing for structural materials with better tolerance and resistance, knowledge of the origins of its most weakening defects are critically needed. Computer simulations and predictive models for material design rely on understanding how irradiation growth begins.

After exposure to neutron irradiation, a large number of vacancies and interstitial atoms are produced, which further evolve into dislocation loops, of which there are two kinds, <a> and <c> loops[1–14]. The <a> and <c> type loops are widely observed in irradiated pure Zr and its alloys and these defects are related to the characteristic c-axis contraction and a-axis expansion seen in irradiated Zr crystals[10,11]. Identifying the formation and growth mechanisms of <a> and <c> loops in Zr is important for understanding and controlling irradiation-induced anisotropic dimension change and minimizing degradation in mechanical properties under irradiation.

The partitioning of <a> interstitial loops on the prismatic planes and <c> vacancy loops on the basal planes is the main driver for anisotropic irradiation growth[15]. The <a> loops, whether interstitial- or vacancy-type loops, lie predominantly on the prismatic plane and have a Burgers vector of $\vec{b} = \frac{1}{3}<11\bar{2}0>$. Almost all <a> dislocation loops have interstitial character at irradiation temperatures below 300 °C[2–4]. Under neutron, proton or electron irradiation, <a> loops tend to align in rows parallel to the basal plane[3–6]. All <c> component loops are vacancy-type loops that lie within the basal plane[6–8]. Significantly, the diameter of the <c> loops observed in experiments, are all larger than 10 nm, which is much larger than the diameter of the <a> loops[14–28]. The absence of vacancy <c> loops less than 10 nm on the basal planes and the presence of only interstitial <a> loops on the prismatic plane cannot explain the observed contraction along the c-axis of the irradiation growth phenomenon[1,11].

Being no smaller than 10 nm has suggested that the <c> loops require an incubation period in which point defects must cluster to a critical size before a <c> loop can be produced. The possible mechanisms for <c> loop formation have received much attention, and over the years, several hypotheses have been proposed[5,29–32]. The formation of the <c> loops has been associated with the coalescence and subsequent collapse of <a> vacancy dislocation loops[5]. However, recent studies indicate that a majority of the vacancy <a> loops lying along the basal plane are randomly and widely distributed in different directions, substantially lowering the odds for coalescence[29]. Alternatively, <c> loops have been thought to be induced by the interaction between a cascade and a dense cluster of pre-existing <a> vacancy loops in the basal plane[29]. The problem with the two aforementioned mechanisms is that they would require a high density of pre-existing <a> vacancy dislocation loops ($\sim 10^{22}$ m$^{-3}$) to match the measured amounts of <c> loops. However, only a few experiments have reached such a high density of <a> dislocation loops[2,5]. Further, vacancy <a> loops develop only under high-temperature irradiation. They can make up to 50% of the <a> loops at temperatures between 400 °C to 450 °C[3] and up to 70% at irradiation temperatures >450 °C[1]. From atomic-scale modeling, a mechanism for <c> loop generation based on the collapse of irradiation-produced pyramids has been proposed[30]. Unlike the prior proposals, the formation of <c> loops in this case would be independent of the formation of <a> vacancy loops, and would instead require the collapse of a stacking fault pyramid that has grown beyond a critical size[30–32]. However, to date, these irradiation-induced pyramidal defects in Zr have not been observed.

In this work, we perform gentle helium implantation on as-annealed high-purity Zr (see composition in Supplementary Table 1) in order to capture the early stages of irradiation defect development before <c> loops have had a chance to fully form. With transmission electron microscope (TEM) analysis, we observe profuse formation of triangle-shaped vacancy platelets (TVPs) on the basal plane, defects that have not been reported before. Based on energetics arguments and the effect of trace elements, we propose a mechanism by which TVP defects can directly transform into <c> loops. The mechanism provides a plausible, experimentally supported explanation for the formation of <c> loops.

## Results

**Dislocation loops**. Figure 1 shows the variation of displacement per atom (dpa) and helium concentration with depth ranging from 0 to 200 nm. This simulation was performed using the Stopping and Ranges of Ions in Matter (SRIM) software with full damage cascades mode and displacement energy of 40 eV[33,34]. After irradiating the TEM foil, the concentration of helium in the 200 nm foil is zero (highlighted by the right corner insert in Fig. 1) and the radiation damage is only 1 dpa. The helium concentration in a thin foil irradiation is different from a bulk sample implantation because the former lacks recoiled atoms, see Supplementary Fig. 1 for details. There are few dislocations in the Zr sample before irradiation (Fig. 1a). Figure 1b, c show the defects formed in the irradiated sample. The thickness at the edge of the TEM foil is <50 nm. No oxide layer was observed at the edge of the foils after irradiation. A high density of ordered nanovoids formed and aligned themselves within the basal plane (Fig. 1b). For the region with thicknesses ranging from 50 to 200 nm, profuse dislocation loops (Fig. 1c) and triangle-shaped defects (Fig. 1d) are observed. The formation of profuse nanovoids in the edge region is related to the clustering of vacancies[35].

Figure 2 shows some key characteristics of the dislocation loops observed in Zr irradiated at 350 °C and 400 °C. All images were taken under strict two-beam conditions. The Burgers vectors of the dislocation loops were determined according to the invisibility criterion (see details in Supplementary Table 2 and Supplementary Fig. 2) and the "inside and outside" diffraction contrast method[3]. Almost all dislocation loops have contrast under $\vec{g} = 1\bar{2}10$ (see Supplementary Fig. 2) and show no contrast with $\vec{g} = 0002$ (Fig. 2a), indicating that most of them are <a> loops. The average size of <a> dislocation loops in irradiated Zr at 350 °C is 5 nm. According to Supplementary Table 2, the dislocation loops numbered with 1, 2, 3 in Supplementary Fig. 2 are vacancy loops with a Burgers vector of $\frac{1}{3}[\bar{2}110]$ and the dislocation loops numbered with 4, 5 in Supplementary Fig. 2 are interstitial loops with a Burgers vector of $\frac{1}{3}[1\bar{2}10]$. The largest projection contrast of the dislocation loops (i.e., loops numbered with 1, 2, 3) under the $[10\bar{1}0]$ (see Supplementary Fig. 2b) indicates that they lie on the prismatic planes. Many dislocation loops were observed sharing similar characteristics with loops 1 to 3 in Supplementary Fig. 2. Nearly 300 dislocation loops were analyzed and the ratio of interstitial to vacancy loops is plotted in Fig. 2d. Interstitial <a> loops occupy 90% of the loops in irradiated Zr at 350 °C. Only one <c> loop was identified in this area, as marked in Fig. 2a. The size of the <c> loop is around 70 nm with a habit plane of (0001) $_{Zr}$.

Radiation defects in irradiated Zr at 400 °C were also analyzed in a similar way (see Supplementary Fig. 3). A large number of coarsened <a> dislocation loops were observed. The fraction of vacancy <a> loops increases from 10 to 30%. Supplementary

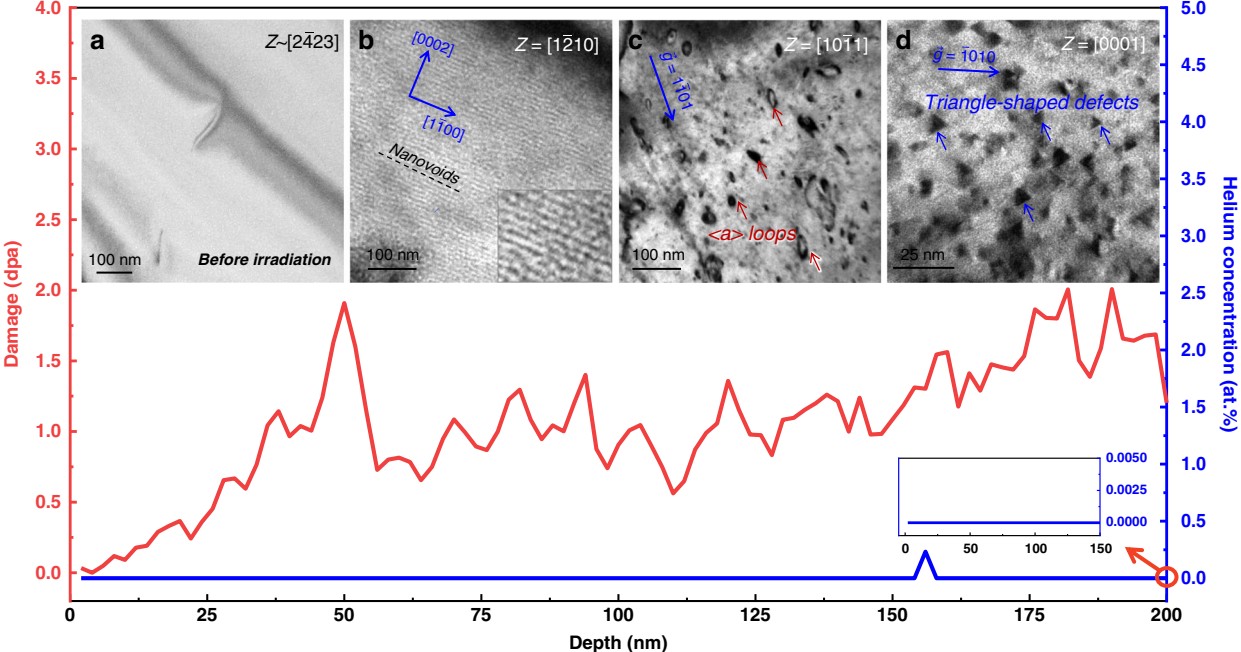

**Fig. 1 The variation in radiation damage and helium concentration in the thin foil Zr. a** TEM image of the as-received non-irradiated Zr. **b** Voids are produced along the basal plane at depths ranging from 0 to 50 nm. The insert in **a** shows an enlarged image of the ordered voids. **c** and **d** Dislocation loops and triangle-shaped defects produced at depths ranging from 50 to 200 nm. The insert shows that the helium concentration is zero.

Fig. 3c demonstrates an example of the coalescence of two dislocation loops observed at 400 °C. The number density of vacancy <a> loop increases at 400 °C. Figure 2d presents the fraction of interstitial and vacancy <a> loops formed at 400 °C. These results indicate more vacancy loops are produced at 400 °C than at 350 °C. We find that most of the <a> dislocation loops are arranged randomly, as shown in Supplementary Figs. 2 and 3. When changing the zone axis to the [1-210] zone axis, some interstitial <a> loops are found aligned in a row, parallel to the prismatic plane, as shown in Supplementary Fig. 2f, similar to the observation previously reported in refs. [9,36], which is related to the anisotropy diffusion of point defects. Under a two-beam image condition, most of the larger <a> vacancy loops have an elliptical shape with the long axis parallel to the [0001] (see Supplementary Fig. 3d). This characteristic elliptical shape is induced by both the elastic anisotropy of Zr and the preferential two-dimensional (2D) interstitial diffusion within the basal plane[37,38]. The number density of dislocation <a> loops is about to $2.6 \times 10^{15}$ m$^{-2}$ at 350 °C and $3.4 \times 10^{15}$ m$^{-2}$ at 400 °C, respectively. The density of vacancy <a> loops at both temperatures is less than $1 \times 10^{15}$ m$^{-2}$, which are insufficient to produce <c> dislocation loops by mechanism of coalescence and collapse[31]. Notably, profuse tiny <c> dislocation loops were also detected in the sample irradiated at 400 °C, as shown in Fig. 2b. The sizes of the <c> loop in this sample range from 20 to 25 nm, while the number density increases significantly compared that under irradiation at 350 °C. A large number of <c> loops also formed in the basal plane in Kr$^{2+}$ irradiated Zr at 400 °C, as shown in Fig. 2c. All of the <c> loops display a line contrast with lengths around 80 nm. The radiation damage in Kr$^{2+}$ implanted Zr is 30 dpa, which is about 30 times higher than that resulting from helium irradiation, in agreement with prior observations that the formation of <c> loops requires a critical amount of radiation damage[8,14].

**Triangle-shaped defects**. Figures 1 and 3 and Supplementary Fig. 4 show that, in addition to the high density of <a> loops and

small fraction of <c> loops, under the viewing direction of [0001], profuse amounts of triangle-shaped defects can also be identified. The triangle-shaped defects, in contrast, are distributed randomly on the basal plane, as labeled by the blue arrows in Supplementary Fig. 3a, b. Supplementary Fig. 3c presents an enlarged image of these triangle-shaped defects. They have an equilateral triangle shape with edges along the three prismatic planes of Zr, as indicated by the selected area diffraction pattern in Fig. 3c. The sizes of the triangle-shaped defects lie in the range of 2 nm to 11 nm (see Fig. 3d and Supplementary Fig. 4). These triangle-shaped defects only show contrast under the [0001] viewing direction and no other visible contrast under the [1$\bar{2}$10] zone axis, indicating that they are platelets (2D) instead of three-dimensional (3D) pyramids. With under focused image conditions, most TVPs have a white contrast (see Supplementary Fig. 4), indicating they are vacancies clusters, similar to voids and helium bubbles[6,35,39]. These special triangle platelets form at both irradiation temperatures. Based on the preferential diffusion and clustering of vacancies in basal plane in Zr under irradiation[35,40–42], these defects can be described as irradiation-induced vacancy plates with their three edges lying along the prismatic/pyramidal planes that intersect their basal habit plane. These nanometer-size, basal plane TVPs have not been experimentally observed in Zr. There are likely two reasons for this fact. First, most of the irradiation defects characterization in the past decades are in Zr alloys with large irradiation damage (>3 dpa), in which the early stages of irradiation defect formation would be difficult to be identified[14,25,26]. Second, most of loops characterized (including both <a> and <c>) were viewed along the [1$\bar{2}$10] zone axis[5–9], while basal plane TVPs only show contrast under [0001] zone axis. Nevertheless, similar shaped defects have been reported in electron-irradiated pure Mg, although they were not analyzed in detail, and in some molecular dynamic simulations of irradiated Zr[7,43].

## Discussion

Based on these experimental observations and distinct size ranges of the <c> loops and TVPs, we propose that <c> loops in Zr are a

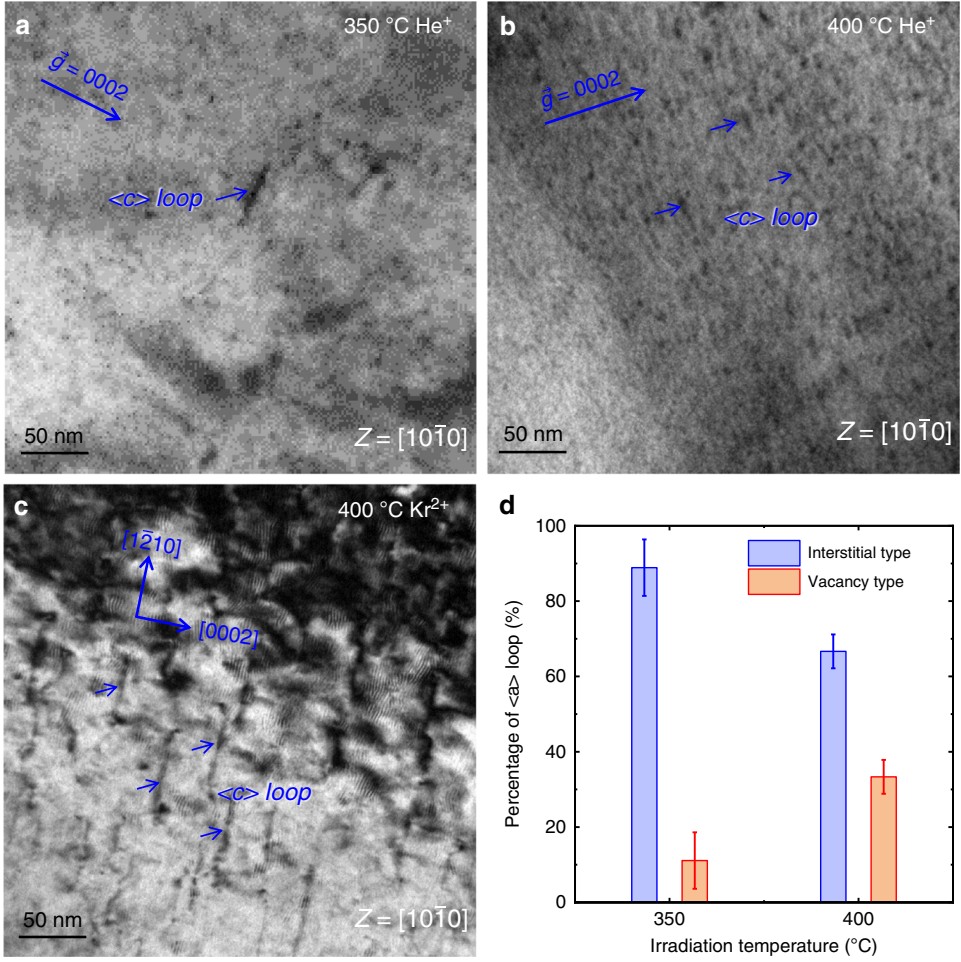

**Fig. 2 Dislocation loops in helium and krypton ion irradiated Zr. a** Radiation defects formed after helium implantation at 350 °C. **b** <c> loops produced by helium ions at 400 °C. **c** <c> loops produced by krypton irradiation at 400 °C. **d** Percentage of <a> interstitial loops and vacancy loops at two irradiation conditions. The statistics involved about 274 loops for 350 °C irradiation and 106 loops for 400 °C irradiation. The error bar is added according to the number of uncertain dislocation loops.

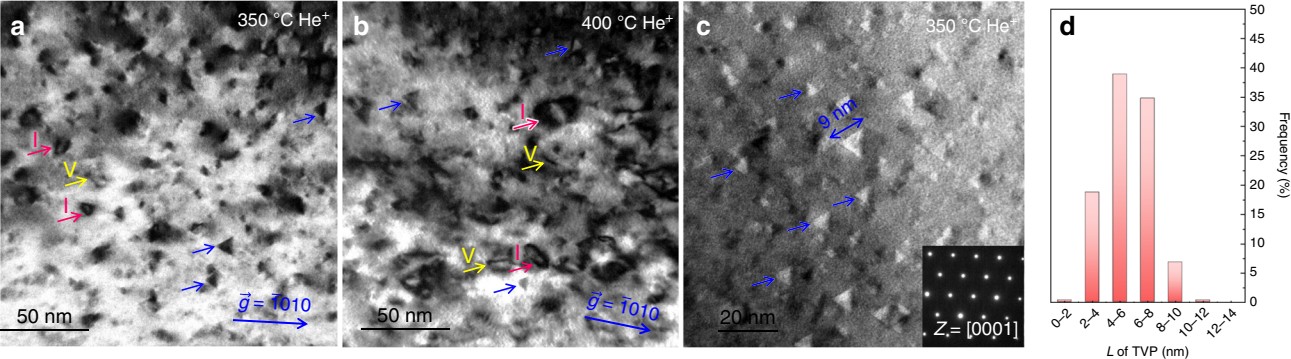

**Fig. 3 Triangle-shaped defects in helium implanted Zr. a** and **b** Profuse triangle-shaped defects and <a> loops (with edge-on orientation) co-exist in Zr. Red arrows label the <a> interstitial loops and yellow arrows mark the <a> vacancy loops. The viewing direction is [0001]. **c** The triangle-shaped defect has equilateral edges along the prismatic planes (defocus of −1000 nm). **d** The size distribution of TVPs. 255 TVPs were involved in the statistics.

consequence of TVPs growing to an unstable size. The <c> loops commonly seen here in pure Zr are similar to those reported in irradiated Zr alloys[15,17,18]. They lie within the basal plane, have a Burgers vector of either $\vec{b} = \frac{1}{2}[0001]$ or $\vec{b} = \frac{1}{6}[20\bar{2}3]$ and are all larger than 13 nm (Fig. 4a)[15–26]. The TVPs discovered here also lie on the basal plane but are distinctly smaller in size

(at most 10.6 nm). If upon reaching a critically large size, these basal plane TVPs are likely to collapse, they could form a <c> dislocation loop. To support this notion, we first compute the elastic energy of circular <c> loop with Burgers vector of $\vec{b} = \frac{1}{2}[0001]$ and compared with the surface energy of the 2D TVPs, as plotted in Fig. 4b.

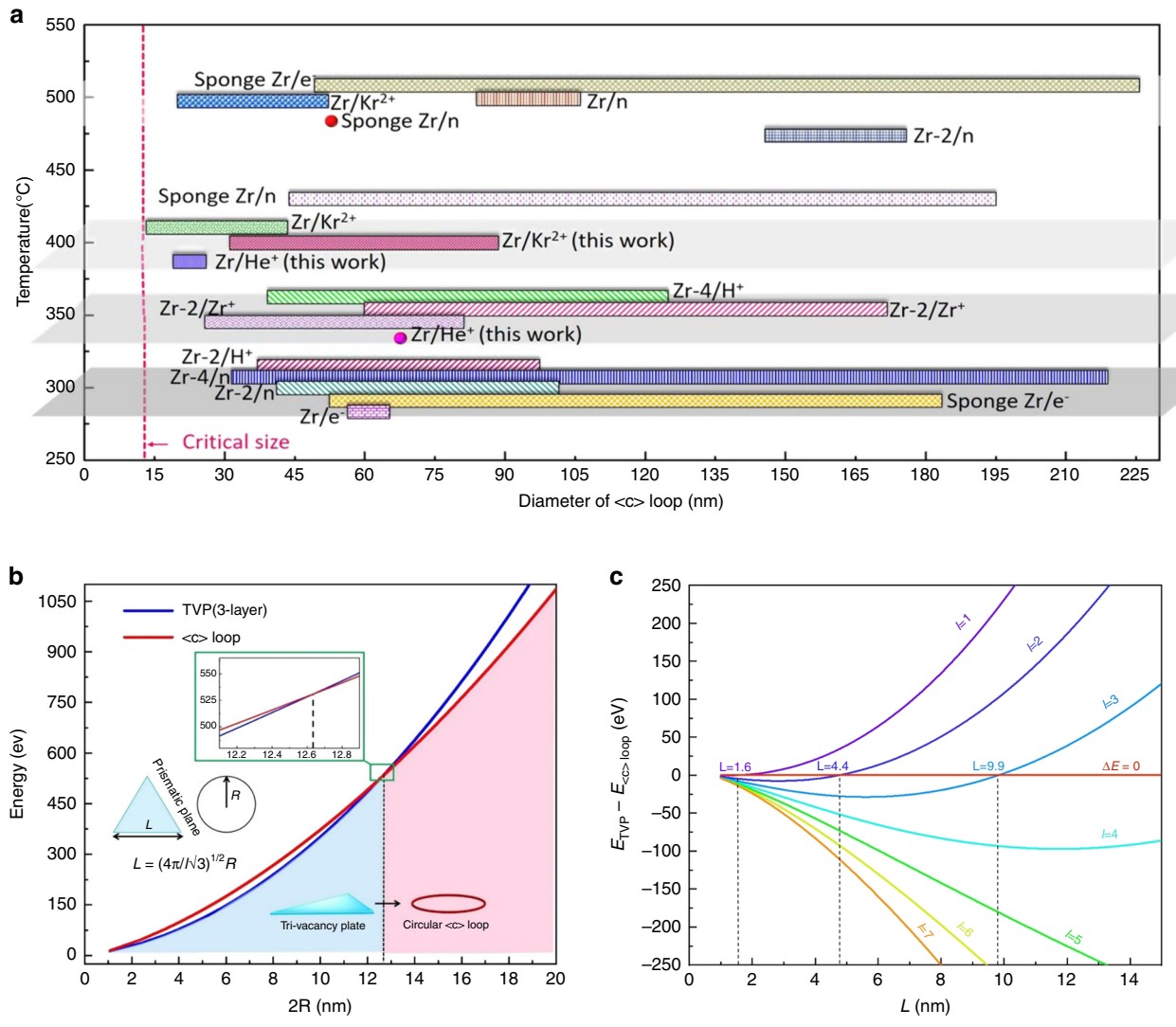

**Fig. 4 Size distribution and energy of <c> loops and triangle-vacancy platelet (TVPs). a** The size of the <c> loops produced at different irradiation temperatures in Zr alloys[14-25]. The pink line marks the lower limit on the size of the <c> loops. **b** Competition of the energy of a <c> loop (blue curve) with a three atomic layer TVP. The inserted images are illustrations of a circular <c> loop and TVP. **c** Highlight of the difference in the energy of a TVP with a <c> loop at various size.

The elastic energy of a circular <c> dislocation loop can be expressed as[44,45]

$$E_{<c>-\text{loop}} = 2\pi R \frac{\mu b^2}{4\pi(1-\vartheta)}\left[\ln\frac{4R}{\rho} - 1\right] + \pi R^2 \gamma_b \qquad (1)$$

where $R$ is the radius of the circular <c> loop, $b = 0.257$ nm is the magnitude of the Burgers vector, $\mu = 34$ GPa is the effective shear modulus of Zr, $\gamma_b = 274$ mJ m$^{-2}$ is the stacking fault energy of basal plane[45], and the $\vartheta = 0.33$ is Poisson's ratio. For an edge dislocation, its core radius can be expressed by $\rho = \frac{b}{2\alpha}\exp(\gamma)$, where $exp(\gamma)$ varies from 1.13 to 1.21, and $\alpha$ is a parameter related to the dislocation core structure, typically ranging from $\alpha = 0.25-2$ in metals[44]. The elastic energy of a <c> loop with $\vec{b} = \frac{1}{6}[20\bar{2}3]$ is close in value to that for a <c> loop with $\vec{b} = \frac{1}{2}[0001]$[45], and in what follows, we use the latter.

In comparison with the loop energy in Eqn. (1), the energy of a 2D TVP can be estimated by considering the energy related to the crystallographic surfaces created by the TVP, which is given by:

$$E_{TVP} = 2A_s\gamma_s + 3lA_l\gamma_l \qquad (2)$$

where $A_s$ is the TVP area on the basal plane and $A_l$ is the area of lateral surface per atomic layer in a vacancy platelet, $l$ is the number of atomic layers of a TVP (which can vary from 1 to 6[46]) and $\gamma_s$ and $\gamma_l$ are the surface energies for the basal and prismatic planes, respectively[47]. For ideally pure Zr, $\gamma_s = \gamma_l = 1.60$ J m$^{-2}$ (ref. [47]). This value is outstandingly high, suggesting that voids or vacancy clusters would not be favored. Yet, several experimental irradiation studies have reported the formation of platelet voids or vacancy defects along the basal plane[16,48]. Actually, even trace amounts of impurities or alloying elements can have a marked effect on surface energies and hence on the stability and clustering behavior of vacancies in Zr[7,49,50]. Simulations show that trace elements, like Fe and Sn, can reduce the vacancy formation energy or stacking fault energy in Zr[49-51]. In the current Zr sample, the impurity elements include Hf, Si, Fe and C, whose concentration are below 20 ppm, as shown in Supplementary Table 1. Because of their low concentration, their effect on defect clustering behavior can be ignored. The oxygen level is slightly higher (140 ppm). However, oxygen has a positive-binding energy with vacancy in Zr, and, thus, cannot stabilize vacancies[51]. In addition, since the helium concentration is zero or very low

(see Fig. 1 and Supplementary Fig. 1), the helium effect can be excluded. Compared with other impurity elements, hydrogen is ubiquitous in Zr and known to play a role in irradiation defect evolution[47,52]. Even minute quantities of hydrogen in Zr can reduce by half or more the surface energy of the basal and/or prismatic planes[47,51].

Figure 4b compares the calculated energies of the <c> loop and TVP from Eqs. (1) and (2) as a function of their size, for a TVP with $l = 3$ atomic layers and considering the prevalent situation with hydrogen impurities, e.g., $\gamma_s = \gamma_l \approx 0.80$ J m$^{-2}$ (ref. [47]). Calculation details can be found in the Supplementary Materials. The comparison finds that the TVP (blue line) has lower system energy than the circular <c> loop (red line) when $2R < 13$ nm. This calculation identifies $L \leq 10$ nm as the stable size regime for the TVP, and once exceeding 10 nm, suggests that the TVP would directly collapse into a more energetically favorable form of a <c> dislocation loop. These critical sizes are consistent with the experimentally observed size ranges for the TVPs and <c> loops in Zr, (see Fig. 3d and Fig. 4a). To examine the influence of TVP layer thickness, Fig. 4c analyzes the difference in energy of a TVP and a <c> dislocation loop. In general, a thicker TVP lowers the formation energy even further and increases the critical size for a stable TVP. Many of the observed TVPs are small with $L = 4$–8 nm, as seen in Fig. 3d. The comparison suggests that the formation of the TVP of size $L$ starts from one or two atomic layers and can gradually thicken into two or three atomic layers, but without a concomitant expansion in size.

As further support, the <c> loop formation mechanism proposed here explains the deficiency of vacancies in the thin TEM foils. During irradiation at these irradiation temperatures, interstitials and vacancies are expected to be produced in equal numbers, especially since the Zr sample was non-textured[17], and accordingly, equal numbers of interstitial and vacancy <a> loops would be expected. While the surface is a sink for both vacancies and interstitials, the interstitials move faster. Consequently, more interstitials than vacancies will be annihilated at the free surface, leaving lower concentrations of interstitials near the surface or in the TEM foils[53]. More vacancy loops than interstitial loops are, therefore, expected in TEM foils compared to the bulk. However, we observe here that the fraction of vacancy <a> loops is no more than 30%, which is less than interstitial <a> loops. Thus, there is the question of where the additional irradiation-induced vacancies went. The fraction of <c> loops is too small to argue that the remaining vacancy <a> loops all coalesced to form <c> loops[29], or that the interaction between the irradiation cascades and a high density of pre-existing <a> vacancy loops accelerated <c> loop formation[5]. While MD simulations have shown that the small size <c> loop can transform from a 3D pyramidal defect, of which the critical size consists of 400 vacancies[30], no such 3D pyramidal defects are observed in our experiment and other studies. Here, we observe a high density of 2D TVPs (one to three atomic layers thick) on the basal plane with sizes no more than 11 nm. The direct transformation of 2D basal vacancy plates into basal <c> loops is a likely explanation for the fate of the remaining irradiation-induced vacancies. However, trace elements, such as hydrogen, etc., can catalyze the formation of TVP due to their effects on surface energy and preferential segregation on basal plane[47,51]. A similar shaped vacancy layer defect was generated in molecular dynamic simulations of high-energy cascades in Zr[43]. Owing to the lack of sufficient numbers of vacancies in a cluster, the vacancy cluster could not immediately collapse into <c> dislocation loops[43]. These vacancy clusters are likely the embryos of TVPs[43].

The concept of vacancy clusters collapsing into dislocation loops has been studied in crystals of cubic symmetry[54–56]. Similar to the formation of Frank loop, we propose that the irradiation-induced vacancies in Zr first accumulate on the basal plane and evolve into 2D TVPs. According to Fig. 4b, the TVP has lower energy than a similar dimension of circular <c> loops when their size is less than 10 nm with the assistance of hydrogen in Zr. The prediction is consistent with observation of TVPs no more than 11 nm (Fig. 3 and Supplementary Fig. 4) and all <c> loops in Zr are larger than 13 nm (Fig. 4a)[28]. In addition, this mechanism provides an explanation for the apparent incubation period for the formation of <c> loops. Finally, the partitioning of irradiation-induced interstitial <a> loops on prismatic plane (Supplementary Fig. 2f) and TVPs or vacancy <c> loops on basal plane (Fig. 2) would produce the anisotropic growth in Zr. These findings can provide insight for managing and reducing the undesirable dimensional and strength changes of hexagonal and related metals under irradiation.

## Methods

**Experimental methods**. High-purity Zr (99.99%) was used in this study, see Supplementary Table 1 for detailed information on the material composition. The high-purity Zr sample was annealed in a high vacuum tube furnace for 3 h with a vacuum higher than $10^{-5}$ Pa. The as-annealed sample has an average grain size of about 100 μm and no strong texture. Thin slices of Zr sample were mechanically ground to a thickness of 30 μm and subsequently electro-polished using a solution of 10% perchloric acid in ethanol at −30 °C. In order to tune the proportion of vacancy dislocation loops, the TEM foils were then irradiated with 400 keV helium (He) ions at 350 °C and 400 °C up to a fluence of $2 \times 10^{17}$ ions m$^{-2}$ with a dose rate of $1.67 \times 10^{13}$ ions m$^{-2}$s$^{-1}$. The irradiation experiment was carried out on the high-energy and medium beam ion implanter produced by the National Electrostatic Corporation of the United States. Before irradiation, the TEM foils were kept in a high vacuum chamber for at least two hours in order to minimize the amount of deposited impurity elements. The oxygen level in the irradiated sample is similar to the sample before irradiation. For comparison, Krypton (Kr) ion (800 keV) irradiation is also performed at 400 °C to a fluence of $5 \times 10^{15}$ ions m$^{-2}$ with a dose rate of $4.6 \times 10^{12}$ ions m$^{-2}$s$^{-1}$. Radiation defects are characterized by using a TEM diffraction contrast technique inside a JEOL 2100F. Conventional "inside and outside" diffraction contrast techniques were adopted to identify the type and crystallography of the loops. The inside/outside technique only works for dislocation loops above a certain critical size. The average size of dislocation loop in this study is around 5 nm, as shown in Supplementary Fig. 5. The number density of dislocation loops is counted based on many TEM images and tiny dislocation loops (i.e., <2 nm) are neglected.

## Data availability

The data that support the findings of this study are available in the paper and supplementary information. Source data are provided with this paper.

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

## Acknowledgements

This work was supported by the National Natural Science Foundation of China (Grant Nos. 51922082, 51971170 and 51942104), the National Key Research and Development Program of China (2017YFB0702301), the 111 Project of China (Grant Number BP2018008) and the Innovation Project of Shaanxi Province (Grant No. 2017KTPT-12).

## Author contributions

W.Z.H. designed the project. S.M.L. performed the experiments under the guidance of W.Z.H. S.M.L., I.J.B. and W.Z.H proposed the theoretical model. All authors discussed, analyzed the results and wrote the manuscript.

## Competing interests

The authors declare no competing interests.
