## [Peer Review File · Nature Communications]

REVIEWER COMMENTS

Reviewer #1 (Remarks to the Author):

A good electron microscopy study of irradiation damage in Zr. It has the potential to improve our understanding of the development of radiation damage structures in Zr. The reasons behind the delay before nucleation of c-loops during irradiation remain a significant unanswered question. The work is fundamentally publishable however there are several factors not considered in the paper. The lack of these factors makes the strong statements that the authors make regarding discovery of a new mechanism highly debatable, at least in the paper as presented. These factors must be discussed in detail before this paper is suitable for publication.

1. "High-purity Zr" - state chemical composition and actual purity as %. Given the sensitivity of Zr damage structure to impurities this is important information. Authors should also state if the material was or was not annealed / stress relieved after grinding prior to experiment.

2. These TFD features look almost identical to triangular defects seen in electron-irradiated Mg on basal planes – see Figure 9 in J. Nuclear Materials, v205, p225 (1993). Note that in Mg especially and also in Zr the loop habit planes are sensitive to material purity – authors need to consider those impurities (especially interstitial impurities); see points 1 and 3.

3. It is also possible that the implantation is introducing interstitial impurities that are affecting the loop habit planes and perhaps character: note that during in-TEM irradiation any vacuum contamination (e.g., trace O) will deposit on the material. This is particularly an issue for Zr given its O gettering capability.

The authors must state how long they left the sample at vacuum before starting their heating/irradiation. If they did not leave it to pump down for many hours before heating, they can expect that trace O from the vacuum will have deposited which would change the impurities present (see J. Appl. Cryst., v50, p1028 (2017) for an extreme case – even if there is not sufficient O present in the TEM column for oxide formation, a great deal of O can dissolve into Zr at these temperatures which would greatly influence defects). This must be considered, and the potential influence discussed.

4. In this context, can the authors comment on why they believe that the TFDs have not been seen before in Zr? There have been several irradiation studies of pure Zr, and the TFDs would be readily observable in the TEM, even 20 years ago (see point 2). It seems strange to me that - if these are a common feature in Zr - they have never been seen. This makes addressing the potential contribution from contamination (points 1, 3), as well as TEM foil-surface / grain orientation (points 6a,6b) and He (point 8) particularly important.

5. Loop analysis: This needs additional clarification, and may need more work. Based on the g.b information presented, the interstitial loops (4,5) in Table S1 could well be in an unsafe orientation, i.e., giving the opposite result. Authors need to specify exactly where they are in orientation space, i.e., which side of (10-11) they are relative to (1-210) for the +(1-10-1) pair. To clarify: the authors are presumably at an orientation close to where $n \cdot B = 0$. In that case they need to know the tilt of the loop with respect to the beam direction and diffracting vectors. I am sceptical of an analysis where the interstitial loops have one Burgers vector and the vacancy loops have another Burgers vector – it is possible the authors have flipped into an unsafe zone for one of the Burgers vectors. When one is at the orientation that the authors are at for that Burgers vector, the Foll and Wilkens method is the best one to apply, not Maher and Eyre which is what they appear to have used (though it is not stated).

6. One would expect that below a few dpa, a-loops are somewhat closer to equal proportions of V and I in character, at least over a sufficiently large average for bulk irradiated materials. It is only after the c-loops start to form that the a-loops become predominantly I, while c-loops are V.

a. The high ratio of I loops observed by the authors could be due to the surface of the TEM foil acting as a preferential sink for vacancies, resulting in the foil material being depleted in vacancies: the strength of the surface sink depends on temperature and flux, see Journal of Nuclear Materials 528 (2020) 151872 for example.

b. In addition, Griffiths (Phil Mag, v45, p613 (1984)) has shown that grains in TEM foils with normals close to prism direction have different V-I ratio than grains with normal close to c-axis, which was attributed to anisotropic diffusion. Was the material rolled? – if it was anything except crystal bar Zr, I expect all the Zr grains observed to have very similar orientation compared to the foil, this would therefore heavily bias the V:I ratio that was determined.

The impact of both 6a) and 6b) mechanisms must be addressed in the paper.

7. Authors' Statement: "some of interstitial  loops are found aligned in a row parallel to the prismatic plane, as shown in Fig. S1f". Alignment of  loops with (11-20) planes has been previously reported (e.g., first by Blake et al, 1974; see discussion in J. Nuclear Materials v159, p190 (1988)); prior studies should be referenced / commented on.

8. From my own SRIM calculations (using same parameters as the authors), I estimate that approx. 1 in 1000 He atoms are in fact implanting in the foil, not zero as stated (we see an example of that in the authors' calculations at 160nm). At these energies / depths implantation is a stochastic process with negligible depth dependence, the authors simply have not run a long enough calculation to see He implant at lower depths; a longer SRIM calculation could be run to show this (and smooth out the calculated damage profile). It is well known that He can act to stabilise V clusters, including in Zr, hence the authors need to discuss the possibility that the TFD features are He stabilised. Based on the fluence, and density of TFDs observed this isn't likely, but it should be quantified not assumed zero.

9. The work in Journal of Nuclear Materials 531 (2020) 151979 suggests that very small planar c-structures in Zr are not stable and will collapse to SFT that seem to have the same crystallographic nature as the authors' TFDs; can the authors comment? The features shown in Fig 3c look near identical to Stacking Fault Tetrahedra (SFTs) commonly observed under irradiation. The evidence in the paper that proves the TFD are purely 2D structures was quite brief; I am not sure it showed there was no small degree of out-of-basal-plane nature - perhaps the authors can quantify the maximum out-of-plane nature based on their viewing conditions, or else provide some additional images to make the 2D nature more clear.

10. Authors' statement: "indicating that the formation of <c> loops requires a critical amount of radiation damage". Since this is well known, please change "indicating that" to "in agreement with prior observations that"

11. Authors' statement: "However, the density of  vacancy loops observed to date have not been sufficiently high to directly produce profuse <c> loops that are more frequently seen." Please provide more references/explanation, I am not sure that this has been conclusively shown.

Reviewer #2 (Remarks to the Author):

The current manuscript presents some experimental evidence on irradiation induced defects in pure Zr. The main finding of the manuscript is the observation (using TEM) of 2D triangular vacancy defects laying on the basal plane of the hexagonal lattice of Zr. The authors hypothesize that these triangular defects are precursors of the c-loops in Zr, i.e c-loops are formed by the collapse of these triangular defects. Their hypothesis is supported by some dislocation and surface energy calculations based on the equations from the literature.

The mechanism of formation of c-loops in Zr alloys is an open debate and there are several proposed theories in the literature. The topic is of great importance in the nuclear materials community since

defects and particularly c-loops play an important role in dimensional stability and mechanical properties of Zr alloys in the nuclear industry.

In the literature, there is at least 1 MD study of α -zirconium which predicts the formation of triangular defects on the basal plane (Wooding, S.J.; Howe, L.M.; Gao, F.; Calder, A.M.; Bacon, D.J. A molecular dynamics study of high-energy displacement cascades in α -zirconium. J. Nucl. Mater. 1998, 254, 191–200). However, to my knowledge there is no experimental observation of such defects in the literature. So in that respect the work can be regarded as novel. I believe the manuscript can benefit from discussing the above-mentioned MD study in detail and comparing their simulations with the experimental findings in the present manuscript.

Overall, the relevant literature has been appropriately cited and discussed.

The manuscript is well-structured and well-written. It was an enjoyable read.

Based on the above I recommend to accept the manuscript , with minor editorial revisions highlighted and commented in the text.

Reviewer #3 (Remarks to the Author):

The authors report the first TEM observation of small triangular platelet defects in irradiated pure Zr. It includes important implications for the unsolved problem of c-loop incubation in Zr and deserve the publication in any scientific papers. Moreover, the manuscript is well organized and written. However, several issues should be clarified before the publication, because of the speciality of Nature Comm.

1. Comments on Abstract

Abstract is too technical, although it is scientifically valuable. The authors should attract wide audience over not only the specialized field but also other fields.

The significance of the results onto whole of materials science should be briefly mentioned.

2. Comments on introductory part

The weakness of introduction part is the same as Abstract.

More wide introduction is required to meet the criteria of Nature Comm.

3. Comments on results and discussion part

3.1, In Fig.3d, the axis labels are too small compared to the figure area.

3.2, In Fig.4, characters written inside the figures are too small compared to the figure area.

3.3, In page 7, line 160, I found "??".

4. Comments on visibility

Simple vacancy platelet does not induce significant strain field around it, and TEM observation should be difficult.

However, if hydrogen atoms are segregated on the platelet surface, as authors assume, and two basal surfaces partly collapse in such a way that hydrogen atoms "glue" two basal surfaces, it may be visible by TEM. In that case, when the cluster completely collapses, hydrogen atoms should move into the bulk interstitial spaces and their energy increases.

Stability analysis of clusters/loops should take this effect into account.

5. Comments on vacancy or interstitial

Molecular dynamics studies[1,2] show that SIA clusters in Zr sometimes have form of triangular platelets.

Authors should logically exclude the possibility of SIA clusters.

References:

[1]Voskoboinikov, R. E., Osetsky, Y. N., & Bacon, D. J. (2006). Nuclear Instruments and Methods in Physics Research Section B:

Beam Interactions with Materials and Atoms, 242(1-2), 530-533.

[2]Voskoboinikov, R. E., Osetsky, Y. N., & Bacon, D. J. (2005). Materials Science and Engineering: A, 400, 54-58.

Reply to comments

For reviewer #1

1. *"A good electron microscopy study of irradiation damage in Zr. It has the potential to improve our understanding of the development of radiation damage structures in Zr. The reasons behind the delay before nucleation of c-loops during irradiation remain a significant unanswered question. The work is fundamentally publishable"*

Reply: Thank you for the time and effort you have spent reviewing our manuscript. We also appreciate the kind words on this research study.

2. *"however there are several factors not considered in the paper. The lack of these factors makes the strong statements that the authors make regarding discovery of a new mechanism highly debatable, at least in the paper as presented. These factors must be discussed in detail before this paper is suitable for publication."*

Reply: Thank you for these valuable comments and suggestions. We have revised our manuscript accordingly. Point-by-point replies to your comments are listed below.

3. *"High-purity Zr"-state chemical composition and actual purity as %. Given the sensitivity of Zr damage structure to impurities this is important information. Authors should also state if the material was or was not annealed / stress relieved after grinding prior to experiment.*

Reply: We agree that information needs to be provided and thank you for your suggestion. The zirconium used in this work is high purity material with a purity of 99.99%. In response, we provide the detailed impurity compositions of the as-received material in Table S1 in the supplementary material. Regarding the second comment, the material was annealed in a high vacuum tube furnace (vacuum better than 10^{-5} Pa) before grinding. After grinding into thin foils, the samples were electro-polished, which can eliminate the grinding-induced stress layers. In response, we have added an image (see revised figure 1) showing the microstructures of the as-prepared sample before ion irradiation.

4. *"These TFD features look almost identical to triangular defects seen in electron-irradiated Mg on basal planes – see Figure 9 in J. Nuclear Materials, v205, p225 (1993). Note that in Mg especially and also in Zr the loop habit planes are sensitive to material purity – authors need to consider those impurities (especially interstitial impurities); see points 1 and 3."*

Reply: We agree this similarity needs to be addressed and appreciate the

reviewer bringing this up. Indeed, the reviewer is correct that the triangle-shaped defects observed in this study is similar to the defects on the basal plane in Mg after electron beam irradiation (*J. Nucl. Mater.*, v205, p225 (1993)). However, in this paper, there are no detailed descriptions and discussions on the nature and the formation mechanism of these defects. Because of their relevancy to the current work, we have cited them in the revised manuscript.

We agree that the trace impurity elements have a significant effect of the formation of irradiation defects and this issue needs to be discussed in the revision. The impurity elements involved in this material includes Hf, Si, Fe and C. But we have assessed that their contents are extremely low, being no more than 20 ppm. The concentration of O is 140 ppm (the effect of O on irradiation defects is discussed below), which is slightly higher than the other elements. In response, we have listed the concentration of those elements in the Table S1 in the supplementary information file. Based on our experiments and analysis, we think that the hydrogen will likely play an important role in the formation of TVP defects in Zr. Notably, compared to the other impurity elements, Zr has a high affinity with hydrogen, which are widely existing in the service environment of Zr alloys and cannot be cleaned up completely. We have also added the following discussion to the revision on pages 8 and 9.

“In the current Zr sample, the impurity elements include Hf, Si, Fe and C, whose concentrations are below 20 ppm, as shown in Table S1. Because of their extremely low concentration, their effect on defect clustering behavior can be ignored. The oxygen level is slightly higher (140 ppm). However, oxygen has a positive binding energy with vacancy in Zr, and thus, cannot stabilize vacancies [53]. In addition, since the helium concentration is zero or extremely low (see Fig. 1 and Fig. S1), the helium effect can be excluded. Compared with other impurity elements, hydrogen is ubiquitous in Zr and known to play a role in the irradiation defect evolution [47,54]. Even minute quantities of hydrogen in Zr can reduce by half or more the surface energy of the basal and/or prismatic planes [47,53].”

5. *“It is also possible that the implantation is introducing interstitial impurities that are affecting the loop habit planes and perhaps character: note that during in-TEM irradiation any vacuum contamination (e.g., trace O) will deposit on the material. This is particularly an issue for Zr given its O gettering capability. The authors must state how long they left the sample at vacuum before starting their heating/irradiation. If they did not leave it to pump down for many hours before heating, they can expect that trace O from the vacuum will have deposited which would change the impurities present (see J. Appl. Cryst., v50, p1028 (2017) for an extreme case – even if there is not sufficient O present in the TEM column for oxide formation, a great deal of O can dissolve into Zr at these temperatures which would greatly influence defects). This must be considered, and the*

potential influence discussed.”

Reply: We thank the reviewer for these good comments and agree these points require clarification in the revision. Indeed, we have paid careful attention on the effect of oxygen in this study. Before starting the heating and irradiation steps, the sample was left in high vacuum for at least 2 hours for this very reason. We have added this information in the revised manuscript. Please see pages 11 where we have added the following:

“Before irradiation, the TEM foils were kept in a high vacuum chamber for at least two hours in order to minimize the amount of deposited impurity elements.”

Thanks for bringing the paper “*J. Appl. Cryst., v50, p1028 (2017)*”, to our attention. We note that in the paper you suggested, the diffusion of oxygen and the formation of oxides occurred at a higher temperature (700°C) than applied here. Consequently, in that work, oxide laths formed at the interface between the oxide layer and the matrix and thus, may be considered an extreme case. In our experiment, the irradiation performed at 400°C and at this temperature, oxygen cannot infiltrate into the sample because of the passive oxide layer (still effective at 400°C) on the sample surface. In addition, no oxide layer was observed at the edge of the foils after irradiation (see Fig. 1). Therefore, the high temperature irradiation cannot introduce additional oxygen. The oxygen levels in the irradiated sample is similar to the sample before irradiation. In response, we have made this point on page 4 and page 12, as follows:

“No oxide layer was observed at the edge of the foils after irradiation.”

“The oxygen level in the irradiated sample is similar to the sample before irradiation.”

Moreover, the DFT calculations show that the binding energy of oxygen and vacancy is positive in Zr (*J. Nucl. Mater. v445, p241 (2014)*). This means that the oxygen impurity cannot stabilize vacancies. Oxygen is slightly more stable as interstitial atom than in a vacancy. This is related to the fact that an oxygen atom is too small to fill a vacancy in a Zr lattice while the interstitial position offers favorable Zr-O bond distances (see discussion in JNM, 2014). Taken together, the oxygen impurity is not the factor promote the formation of TVPs. We agree that this is an important point to mention for the benefit of the readers. We have added the following point in the revision on page 8.

“The oxygen level is slightly higher (140 ppm). However, oxygen has a positive binding energy with vacancy in Zr, and, thus, cannot stabilize vacancies [53]. ”

6. *“In this context, can the authors comment on why they believe that the TFDs*

have not been seen before in Zr? There have been several irradiation studies of pure Zr, and the TFDs would be readily observable in the TEM, even 20 years ago (see point 2). It seems strange to me that - if these are a common feature in Zr - they have never been seen. This makes addressing the potential contribution from contamination (points 1, 3), as well as TEM foil-surface / grain orientation (points 6a,6b) and He (point 8) particularly important."

Reply: This is a great question and one that needs to be addressed at the outset of the revised manuscript. Firstly, to the best of our knowledge of the relevant investigations in the literature to date, there has been no report on the formation of TVPs in Zr. The triangle-shaped defects in Mg (see "*J. Nucl. Mater. V205, p225, (1993)*") appear similar to the TVPs in this study. However, these defects were not analyzed and explained in detail.

We think there are three likely reasons for the lack of prior reports of these TVPs in irradiated Zr.

First, the observation direction under TEM is important. In past decades, most works characterized the $\langle c \rangle$ component dislocation loops from the $\langle 1-210 \rangle$ zone axis, along which the c-loops only show a line contrast (see "*Acta Mater., v165, p603 (2019), J. Nucl. Mater., v514, p358 (2019), Mater. Design, v161, p147 (2019), J. Nucl. Sci. Technol., v55, p1212 (2018), J. Nucl. Mater., v441, p138 (2013), J. Nucl. Mater., v425, p76 (2012) and Philos. Mag., v88, p649 (2006)*"). According to our calculation, the TVP defects only have a few atomic layers in thickness, which is difficult to identify under $\langle 1-210 \rangle$ zone axis under a lower magnification, even under in-situ irradiation. As shown in this study, the TVPs only show contrast under or near $[0001]$ zone axis.

Second, TVPs are the earlier stage of irradiation defect formation, compare to prior works. Hence, the small irradiation damage stage is turning out to be a precondition for effective observation of the TVPs. There are several studies to characterize radiation defects in Zr under $[0001]$ zone axis, but their samples underwent high levels of irradiation damage. For example, the typical irradiation damage amount in most studies is about 7-50 dpa, and under these intense conditions, many $\langle c \rangle$ loops have already formed in these samples. The high density of $\langle c \rangle$ loops likely mask the early-stage irradiation defects, such as the TVPs identified in this study. See "*Acta Mater., v130, p69 (2017), J. Nucl. Mater., v488, p33 (2017), J. Nucl. Mater., v491, p232 (2017) and J. Nucl. Mater., v433, p95 (2013)*". Unlike prior works, in this study, we designed a gentle-ion irradiation condition with radiation damage of only 1 dpa. Under the current irradiation situation, the earlier stage of irradiation defects in Zr can be readily captured.

Third, the high purity of the Zr likely play a role. Many earlier studies have

chosen Zr alloys or low impurity Zr (99.9%) to focus their study (see “*J. Nucl. Mater.*, v115, p323, (1983), *Philos. Mag. A*, v49, p613 (1984), *J. Nucl. Mater.*, v137, p185 (1986) and *J. Nucl. Mater.*, v150, p159 (1987)”). Impurity or alloy elements, such as Fe, Sn, Cr, Ni and Nb can stabilize vacancies and dislocation loops in Zr (see “*J. Nucl. Mater.*, v445, p241 (2014) and *Phys. Rev. Mater.*, v3, 043602 (2019)”). Those alloy elements likely catalyze the formation of <c> dislocation loops, which make the capture of TVP defects difficult. On the contrary, our sample is of high purity (>99.99%, also see Table S1), which is similar to Mg with 99.999% purity in the aforementioned study (“*J. Nucl. Mater.*, v205, p225 (1993)”). We speculate that this could be one reason why that Mg study and the present Zr study observed a similar phenomenon after irradiation.

In the revision, we outline these points in the following ways:

“These nanometer-size, basal plane TVPs have not been experimentally observed in Zr. There are likely two reasons for this fact. First, most of the irradiation defects characterization in the past decades are in Zr alloys with large irradiation damage (>3 dpa), in which the early stages of irradiation defect formation would be difficult to be identified [14,25-26]. Second, most of loops characterized (including both  and <c>) were viewed along the $[1\bar{2}10]$ zone axis [5-9], while basal plane TVPs only show contrast under $[0001]$ zone axis. Nevertheless, similar shaped defects have been reported in electron-irradiated pure Mg, although they were not analyzed in detail, and in some molecular dynamic simulations of irradiated Zr [7,43].”

7. *“Loop analysis: This needs additional clarification, and may need more work. Based on the g.b information presented, the interstitial loops (4,5) in Table S1 could well be in an unsafe orientation, i.e., giving the opposite result. Authors need to specify exactly where they are in orientation space, i.e., which side of (10-11) they are relative to (1-210) for the +(1-10-1) pair. To clarify: the authors are presumably at an orientation close to where $n.B=0$. In that case they need to know the tilt of the loop with respect to the beam direction and diffracting vectors. I am sceptical of an analysis where the interstitial loops have one Burgers vector and the vacancy loops have another Burgers vector – it is possible the authors have flipped into an unsafe zone for one of the Burgers vectors. When one is at the orientation that the authors are at for that Burgers vector, the Foll and Wilkens method is the best one to apply, not Maher and Eyre which is what they appear to have used (though it is not stated).”*

Reply: Thank you for your suggestion. We agree that our loop analysis could use more clarification. In fact, we also used the Foll and Wilkens method to analyze our results. We did not state it but agree that we should have. In the revision we have now noted this on page 4.

We analyze the nature of irradiation loops using the “inside and outside” diffraction contrast method according to “A. Jostsons et al. *The nature of dislocation loops in neutron irradiated zirconium. J. Nucl. Mater. v66, p236, (1977)*”. The deviation vector (\vec{s}) is positive so that we can get a better contrast of dislocation loops. We also ensure that the loops (4, 5) are in a safe position, as the enlarged images (a-e) show below.

When judging the nature of the dislocation loops and determining whether it is in a safe orientation, it depends on the habit plane of loops and its normal direction. In our study, the sample holder cannot tilt to an edge-on position of the loops, as Jostson et al did (“*J. Nucl. Mater. v66, p236, (1977)*”). Consequently, we observe the relative area of the dislocation loops by tilting the zone axis. In general, the closer the zone axis is to the habit plane, the larger the projection area of dislocation loop. When tilting from $[10\bar{1}0]$ to $[\bar{2}1\bar{1}0]$, we find that the area of dislocation loops 4 and 5 tend to become larger. This change means these loops are on the $(1\bar{1}00)$ prismatic plane instead of the $(01\bar{1}0)$ prismatic plane. Because if they were on the $(01\bar{1}0)$ plane, then the dislocation loops should have shown an edge-on contrast. The position of their Burgers vectors and loop normals are illustrated in Fig. f below. The loops marked in Fig. S2 (1,2,3,4,5) are simply just some examples to demonstrate our analysis. Actually, there are many other dislocations loops having the same Burgers vector as the loops 1, 2 and 3, but they are interstitial loops, which are not marked in the figures.

In the revision, according to the above description, we added more information on the loop analysis. It is provided below and can be found on pages 4-5.

“The Burgers vectors of the dislocation loops were determined according to the invisibility criterion (see details in Table S2 and Fig. S2) and the “inside and outside” diffraction contrast method [3].”

“The largest projection contrast of the dislocation loops (i.e., loops numbered with 1,2,3) under the $[10\bar{1}0]$ (Fig. S2b) indicates that they lie on the prismatic planes. Many dislocation loops were observed sharing similar characteristics with loops 1 to 3 in Fig. S2.”

8.1 *“One would expect that below a few dpa, a-loops are somewhat closer to equal proportions of V and I in character, at least over a sufficiently large average for bulk irradiated materials. It is only after the c-loops start to form that the a-loops become predominantly I, while c-loops are V.”*

Reply: We agree that the proportions of V and I is equal in bulk irradiated materials, in regions below a few dpa.

8.2 *“a. The high ratio of I loops observed by the authors could be due to the surface of the TEM foil acting as a preferential sink for vacancies, resulting in the foil material being depleted in vacancies: the strength of the surface sink depends on temperature and flux, see Journal of Nuclear Materials 528 (2020) 151872 for example.”*

Reply: We agree that the surface has some effect on the dynamics of irradiation defects. As mentioned in “*J. Nucl. Mater.* v528, 151872 (2020) and *Mater. Charact.* v151, p175 (2019)”, the surface effect should be taken into consideration for defect evolution in thin foils. Vacancies are mobile at 400°C. It is our understanding that the free surface acts as sink for both interstitials and vacancies. However, in general, interstitials move faster than vacancies. Therefore, more interstitials are likely to migrate to the surface and produce a higher concentration of vacancies in the thin foils. According to this process, we should see more vacancy dislocation loops in the thin film sample. However, we identified more interstitial dislocation loops (see Fig. 3d). Based on this fact and the equal number of V and I under irradiation, we think that a large fraction of vacancies is not in the form of dislocation loops, but likely stored in the materials as other types of defects. In the revision, we clarified this point on page 10, as follows:

“During irradiation at these irradiation temperatures, interstitials and vacancies are expected to be produced in equal numbers, especially since the Zr sample was non-textured [17], and accordingly, equal numbers of interstitial and vacancy  loops would be expected. While the surface is a sink for both vacancies and interstitials, the interstitials move faster. Consequently, more interstitials than vacancies will be annihilated at the free surface, leaving lower

concentrations of interstitials near the surface or in the TEM foils [54]. More vacancy loops than interstitial loops are, therefore, expected in TEM foils compared to the bulk.”

- 8.3 *“In addition, Griffiths (Phil Mag, v45, p613 (1984)) has shown that grains in TEM foils with normals close to prism direction have different V:I ratio than grains with normal close to c-axis, which was attributed to anisotropic diffusion. Was the material rolled? – if it was anything except crystal bar Zr, I expect all the Zr grains observed to have very similar orientation compared to the foil, this would therefore heavily bias the V:I ratio that was determined. The impact of both 6a) and 6b) mechanisms must be addressed in the paper.”*

Reply: We agree that the texture has an effect on the diffusion and distribution of defects in Zr. In general, the diffusion of point defects is much faster along a-axis than along c-axis, and the diffusion of interstitials is much faster than vacancies. We cited the reference you suggested. In order to identify the TVP defects, we observe the sample along the c-axis, which is also the irradiation direction. Hence, as least in those grains, the V:I ratio is close to 1. In this study, the Zr sample underwent a long time annealing in a high vacuum tube furnace and therefore, the texture is relatively random. This is consistent with our observations based on our extensive TEM observation. Therefore, we suspect that the texture of the sample does not have a big effect.

We have added text to address these points in the revised manuscript on page 12. The added portion reads: “The high-purity Zr sample was annealed in a high vacuum tube furnace for 3 h with a vacuum higher than 10^{-5} Pa. The as-annealed sample has an average grain size of about 100 μm and no strong texture.”

9. *“Authors’ Statement: “some of interstitial loops are found aligned in a row parallel to the prismatic plane, as shown in Fig. S1f”. Alignment of loops with (11-20) planes has been previously reported (e.g., first by Blake et al, 1974; see discussion in J. Nuclear Materials v159, p190 (1988)); prior studies should be referenced / commented on.”*

Reply: Thank you for your suggestion. We have added this reference in the revised manuscript.

10. *“From my own SRIM calculations (using same parameters as the authors), I estimate that approx. 1 in 1000 He atoms are in fact implanting in the foil, not zero as stated (we see an example of that in the authors’ calculations at 160nm). At these energies / depths implantation is a stochastic process with negligible depth dependence, the authors simply have not run a long enough calculation to see He implant at lower depths; a longer SRIM calculation could be run to show this (and smooth out the calculated damage profile). It is well known that He can*

act to stabilise V clusters, including in Zr, hence the authors need to discuss the possibility that the TFD features are He stabilised. Based on the fluence, and density of TFDs observed this isn't likely, but it should be quantified not assumed zero.”

Reply: We conducted the calculation again using SRIM with a total number of 99999 helium ions implantation into Zr foils with different thicknesses as well as in bulk Zr. We ensured that the calculation run time was sufficiently long to allow for He implantation should it occur. Except for the foil with 50 nm thickness, the foils with thickness of 150 nm and 200 nm all show a small helium concentration peak, which means just one or none helium in those thin foils sample. The small peak is likely just a statistic result of SRIM calculation. The results are provided in the figures below.

The helium concentrations in bulk Zr are completely different. For bulk samples, helium ions will be scattered at large angles after implantation (that is, the scattering angle is greater than 150 degrees). In that case, helium ions bounced back to the shallow region in the sample, thus leading to higher helium concentrations in the 0 to 200 nm range in bulk implantation. For the thin film samples, however, because of lacking of recoil atoms in the back, the helium concentration is extremely low. Please see the figure below. According to the calculation, although there exists a small peak, the helium concentration is zero or extremely low, which is too low to stabilize such a large number of vacancy defects as observed in the experiment. Therefore, the helium effect can be excluded in this study. We have added text to address these points in the revised manuscript on page 4 and page 9 and the figures below in the supplementary information file.

11. "The work in *Journal of Nuclear Materials* 531 (2020) 151979 suggests that very small planar c-structures in Zr are not stable and will collapse to SFT that seem to have the same crystallographic nature as the authors' TFDs; can the authors comment? The features shown in Fig 3c look near identical to Stacking Fault Tetrahedra (SFTs) commonly observed under irradiation. The evidence in the paper that proves the TFD are purely 2D structures was quite brief; I am not sure it showed there was no small degree of out-of-basal-plane nature - perhaps the authors can quantify the maximum out-of-plane nature based on their viewing conditions, or else provide some additional images to make the 2D nature more clear."

Reply: Thank you for your advice. This article you mentioned is similar to the reference "*Acta Mater.* v179, p93, (2019)", both of which discuss the formation of c-loop and related to our current research. We have cited the references in the revised manuscript. According to their simulations, vacancies cluster into pyramid defects in Zr; however, there are no experimental observation supporting these claims.

The nature of the 2D TVP is judged based on the following reason. First, when we tilt the zone axis to $[1\bar{2}10]$ or nearby (basal plane on edge on position), we cannot identify any triangle or trapezoid contrast. The TVPs only show contrast under the c-axis observation direction. This is different from the contrast of stacking fault tetrahedrons (SFT). Second, the high-resolution TEM image of SFTs in copper and TVPs in our experiment show different contrast, see figure below. For example, SFT has deeper contrast on one side and shallower contrast on the other side. If TVP has a 3D structure like SFT, its contrast should be uneven. However, the TVP has relative homogeneous contrast within the whole triangle region. Therefore, we think the TVP is a 2D platelet defect on basal plane.

The vacancy nature of TVPs is judged based on the following reasons. First, we observed more interstitial loops than the vacancy loops, because of the production of V and I should be equal, thus a fraction of vacancies should store in other form of defects other than dislocation loops, such TVPs. The mobility of vacancies is high at 400°C, thus vacancies are not likely still frozen in the lattice in Zr. Second, we used under focused imaging condition to study these TVPs, and most of them have a white contrast (Fig. S4), which is a common feature for the vacancy clusters, such as voids or helium bubbles (see figure below). Third, according to the anisotropic diffusion and formation energy of point defects, vacancies prefer to stay on the basal plane, as indicated by the aligning of helium bubbles along the basal plane in Zr (see “*J. Mater. Sci. Technol.* v35, p1466 (2019)”). Therefore, the TVPs are 2D vacancy defects.

12. *Authors' statement: "indicating that the formation of <c> loops requires a critical amount of radiation damage". Since this is well known, please change "indicating that" to "in agreement with prior observations that"*

Reply: Thanks for your suggestion. We have made this change in the revised manuscript.

13. *Authors' statement: "However, the density of vacancy loops observed to date have not been sufficiently high to directly produce profuse <c> loops that are more frequently seen." Please provide more references/explanation, I am not sure that this has been conclusively shown.*

Reply: Thanks for your suggestion. In fact, in some works (“*J. Nucl. Mater.* v61, p123 (1976), *J. Nucl. Mater.* v115, p313 (1983), *J. Nucl. Mater.* v96, p213 (1987), *J. Nucl. Mater.* v150, p169 (1987), *J. Nucl. Mater.* v159, p43 (1988), *J. Nucl. Mater.* v205, p225 (1993), *J. Nucl. Mater.* v423, p170 (2012), *J. Nucl. Mater.* v425, p76 (2012) and *J. Nucl. Mater.* v491, p232 (2017)”), the density of  dislocation loops produced by their irradiation didn’t reach the quantitative order of 10^{22}

m^{-3} , as referred by “*Scripta Mater.* v172, p72 (2019)”. Only a few works (“*J. Nucl. Mater.* v61, p123 (1976), *Acta Mater.* v130, p69 (2017) and *Acta Mater.* v165, p603 (2019)”) reported that the density of $\langle a \rangle$ dislocation loops had reached this quantitative order. We have added some references and modified this expression as follows:

“The problem with the two aforementioned mechanisms is that they would require a high density of pre-existing $\langle a \rangle$ vacancy dislocation loops ($\sim 10^{22} \text{ m}^{-3}$) to match the measured amounts of $\langle c \rangle$ loops. However, only a few experiments have reached such a high density of $\langle a \rangle$ dislocation loops [2,5].”

Questions and suggestions marked in the manuscript file

14. “*Why the vacancy loops on the basal plane have to be less than 10 nm to explain observed expansion along a-axis? As far as I know, there is no claim about loop size in Ref [1] and [11]*”

Reply: According to the experimental observations in [1] and [11], all the identified $\langle c \rangle$ loops have a size larger than 10 nm, and no $\langle c \rangle$ loop with size less than 10 nm were reported. In order to explain irradiation growth, we need to have both of interstitial $\langle a \rangle$ loop and vacancy $\langle c \rangle$ loop. However, in most experiments, there are no $\langle c \rangle$ loops at low irradiation dose, only $\langle a \rangle$ loops on the prismatic plane can be observed. The lack of $\langle c \rangle$ loops in the earlier stage of irradiation cannot explain the c-axis contraction. In order to avoid misunderstanding, we have rewritten this part on page 2.

The revised portion reads: “The absence of vacancy $\langle c \rangle$ loops less than 10 nm on the basal planes and the presence of only interstitial $\langle a \rangle$ loops on the prismatic plane cannot explain the observed contraction along the c-axis of the irradiation growth phenomenon [1-11].”

15. “ *$\langle a \rangle$ loops mostly lie on the prismatic planes. It's has been suggested in the literature that coalescence and collapse of those $\langle a \rangle$ loops on the prismatic planes that are aligned parallel to the trace of the basal plane forms $\langle c \rangle$ loops. See your ref [5]*”

Reply: We understand the comment and thank the reviewer. Although the $\langle a \rangle$ loops on the prismatic plane are arranged in a row along the basal plane, they do not align in the same direction. This expression from Ref [29] may cause ambiguity. We have revised the related statements in the revised manuscript, which reads:

“However, recent studies indicate that a majority of the vacancy  loops that lie along the basal plane are randomly and widely distributed in different directions, substantially lowering the odds for coalescence [29].”

16. *“Was the thickness measured for example with EELS ZLP?”*

Reply: We did not measure the sample thickness using EELS. The sample thickness is judged based on the electron transparency and image contrast. In general, the TEM sample has a thickness of less than 150 nm.

17. *“Would be helpful to include a micrograph of the as-received non-irradiated sample? ”*

Reply: We agree and we have added a micrograph of the as-received non-irradiated sample in revised Fig. 1.

18. *“Would be nice to mention the total number of analyzed loops.”*

Reply: We agree and we have added this information in the revised manuscript.

19. *“There has been reports in the literature about these type of defects; see: 1) Wooding, S.J.; Howe, L.M.; Gao, F.; Calder, A.M.; Bacon, D.J. A molecular dynamics study of high-energy displacement cascades in α -zirconium. J. Nucl. Mater. 1998, 254, 191–200.”*

Reply: Thank you for your suggestion. We have cited these papers and revised the related statements in the revised manuscript.

20. *“Please provide some info about the number of measured TVPs”.*

Reply: We agree and we have added the information to the revised manuscript.

21. *“It would be nice to mention in what instrument/facility the irradiation was done.”*

Reply: We agree and we have added this information in the revised manuscript.

22. *“References should be re-arranged so that they are in order throughout the text. Here after ref [3], it jumps to [10,11]”*

Reply: Thank you for your careful reading. The order of the references has been

re-arranged accordingly.

23. *Marked directly in manuscript. "Fig. S5 is missing" "can be formed" "alloying"*

Reply: Thank you for your suggestion. We have revised the text accordingly in the revised supplementary materials.

For reviewer #2:

“The mechanism of formation of c-loops in Zr alloys is an open debate and there are several proposed theories in the literature. The topic is of great importance in the nuclear materials community since defects and particularly c-loops play an important role in dimensional stability and mechanical properties of Zr alloys in the nuclear industry.

In the literature, there is at least 1 MD study of α -zirconium which predicts the formation of triangulare defects on the basal plane (Wooding, S.J.; Howe, L.M.; Gao, F.; Calder, A.M.; Bacon, D.J. A molecular dynamics study of high-energy displacement cascades in α -zirconium. J. Nucl. Mater. 1998, 254, 191–200). However, to my knowledge there is no experimental observation of such defects in the literature. So in that respect the work can be regarded as novel. I believe the manuscript can benefit from discussing the above-mentioned MD study in detail and comparing their simulations with the experimental findings in the present manuscript.”

Reply: We are grateful for your evaluation, positive comments and valuable recommendation. We have cited the reference in our revised manuscript and compared with our experimental observations. According to the simulations, vacancy clusters have a multi-layer structure, and their projection on the basal plane has a triangular shape. This observation is very similar to the TVPs identified in current experiment. In the revision, we have added the following discussion on page 11.

“A similar shaped vacancy layer defect was generated in molecular dynamic simulations of high energy cascades in Zr [42]. Due to the lack of sufficient numbers of vacancies in a cluster, the vacancy cluster could not immediately collapse into <c> dislocation loops [42]. These vacancies cluster are likely the embryos of TVPs [42].”

For reviewer #3:

1. *“Comments on Abstract*

Abstract is too technical, although it is scientifically valuable. The authors should attract wide audience over not only the specialized field but also other fields. The significance of the results onto whole of materials science should be briefly mentioned.”

Reply: We appreciate very much the reviewer’s comment and agree that this change to the abstract is necessary. Per your suggestion, we have tried our best to improve it. Please see the new version in the revision.

2. *“Comments on introductory part*

The weakness of introduction part is the same as Abstract. More wide introduction is required to meet the criteria of Nature Comm.”

Reply: Again, we are grateful for the important comment and agree that modification of the introduction to reach a broad audience is important. We have attempted to improve it. Please see the new version in the revision.

3. *“Comments on results and discussion part*

- 3.1, In Fig.3d, the axis labels are too small compared to the figure area.*
- 3.2, In Fig.4, characters written inside the figures are too small compared to the figure area.*
- 3.3, In page 7, line 160, I found "??".”*

Reply: Thanks for your suggestion! We have revised those points in our revised manuscript.

4. *“Comments on visibility*

Simple vacancy platelet does not induce significant strain field around it, and TEM observation should be difficult. However, if hydrogen atoms are segregated on the platelet surface, as authors assume, and two basal surfaces partly collapse in such a way that hydrogen atoms "glue" two basal surfaces, it may be visible by TEM. In that case, when the cluster completely collapses, hydrogen atoms should move into the bulk interstitial spaces and their energy increases. Stability analysis of clusters/loops should take this effect into account.”

Reply: Thank you for your discussion. In our work, hydrogen influences the surface energy of basal plane (TVP surface energy) instead of the stacking fault energy of basal plane. The presence of hydrogen reduces the surface energy, decreasing the TVP formation energy, thus TVPs are stable below a critical size. Once the TVPs collapse into c-loops, the effect of hydrogen disappears. The

energy of a c-loop depends on their size and the stacking fault energy of basal plane. However, the stacking fault energy of basal plane is a constant and not affected by the hydrogen.

5. "Comments on vacancy or interstitial

Molecular dynamics studies[1,2] show that SIA clusters in Zr sometimes have form of triangular platelets. Authors should logically exclude the possibility of SIA clusters.

[1]Voskoboinikov, R. E., Osetsky, Y. N., & Bacon, D. J. (2006). *Nuclear Instruments and Methods in Physics Research Section B: Beam Interactions with Materials and Atoms*, 242(1-2), 530-533.

[2]Voskoboynikov, R. E., Osetsky, Y. N., & Bacon, D. Z. (2005). *Materials Science and Engineering: A*, 400, 54-58."

Reply: Thank you for your suggestions and the references. From the experimental point of view, it is really difficult to judge whether the TVPs are SIA cluster or vacancy cluster. The vacancy nature of TVPs is judged based on the following reasons. First, we observed more interstitial loops than the vacancy loops, because of the production of V and I should be equal, thus a fraction of vacancies should store in other form of defects other than dislocation loops, such TVPs. The mobility of vacancies is high at 400°C, thus vacancies are not likely still frozen in the lattice in Zr. Second, we used under focused imaging condition to study these TVPs, and most of them have a white contrast (Fig. S4-see below), which is a common feature for the vacancy clusters, such as voids or helium bubbles. Third, according to the anisotropic diffusion and formation energy of point defects, vacancies prefer to stay on the basal plane, as indicated by the aligning of helium bubbles along the basal plane in Zr (see "*J. Mater. Sci. Technol.* v35, p1466 (2019)"). Therefore, the TVPs are 2D vacancy defects.

REVIEWERS' COMMENTS

Reviewer #1 (Remarks to the Author):

The authors have carefully responded to the reviewers' comments. The paper is now acceptable for publication.

Reviewer #2 (Remarks to the Author):

All the comments were addressed with careful consideration in the revised version of the manuscript. Thank you for the nice work.